# Adaptation and Validation of the Turkish Version of the Brain Fog Scale

**DOI:** 10.3390/ijerph21060774

**Published:** 2024-06-14

**Authors:** Murat Bas, Meryem Kahriman, Cansu Gencalp, Selen Koksal Koseoglu, Ladan Hajhamidiasl

**Affiliations:** Department of Nutrition and Dietetics, Faculty of Health Sciences, Acibadem Mehmet Ali Aydinlar University, Istanbul 34752, Türkiye; meryem.kahriman@acibadem.edu.tr (M.K.); cansu.zirtil@acibadem.edu.tr (C.G.); selen.koksal@acibadem.edu.tr (S.K.K.); ladan.hajhamidi@gmail.com (L.H.)

**Keywords:** brain fog, COVID-19, cognitive function, validation, scale

## Abstract

Brain fog is a condition that is characterized by poor concentration, memory loss, decreased cognitive function, and mental fatigue. Although it is generally known as a long-term COVID-19 symptom, brain fog has also been reported to be caused by many other diseases. Thus, it is necessary to assess this condition in certain populations. This study aimed to evaluate the reliability and validity of the Brain Fog Scale in a Turkish population. We conducted the study in two phases. In a pilot study including 125 participants, we confirmed the suitability of the scale for validity analyses and then conducted exploratory (*n* = 230) and confirmatory factor analyses (*n* = 343). The Cronbach’s alpha value of the 23-item Brain Fog Scale was 0.966. In addition, the 23-item and three-factor structure was confirmed as a result of the analyses. These three factors are mental fatigue, impaired cognitive acuity, and confusion. We also found that participants previously diagnosed with COVID-19 had higher brain fog scores. This finding indicates that brain fog is an important condition that can accompany COVID-19. Furthermore, this validated construct has an acceptable fit and is a valid and useful tool for the Turkish population.

## 1. Introduction

Brain fog is a term used to describe individuals’ experiences when their cognitive functions are not as sharp as usual [1,2]. This condition is characterized by mental fatigue, memory loss, confusion, anxiety, decreased concentration, tiredness, injury, and impaired cognitive functions [2,3,4]. Although brain fog is generally known as a long-term symptom of COVID-19 [5], it can also be caused by other diseases such as chronic fatigue syndrome [6], cancer [7], celiac disease [8], systemic lupus erythematosus [9], hypoparathyroidism [10], and postural tachycardia syndrome [11]. It can also be triggered by factors such as poor nutrition, strenuous physical activity, medications, and menopausal transition [6,12,13].

Brain fog has been a controversial issue, particularly during the COVID-19 period, so much so that the Centers for Disease Control and Prevention considered it to be among the long-term symptoms of COVID-19 [14]. Furthermore, brain fog has been reported as a common complication of COVID-19 [15]. In their meta-analysis of 81 studies, Ceban et al. [16] reported that a significant portion of individuals experienced cognitive impairment and/or persistent fatigue after recovering from COVID-19. Malik et al. [17], in their meta-analysis of 12 studies, reported that post-acute COVID-19 syndrome is associated with permanent symptoms such as fatigue, sleep disorders, and poor mental health. The prevalence of brain fog in COVID-19 and other pathological conditions highlights its impact on individuals’ health and quality of life. Brain fog may also be associated with poor self-perception, a lack of support, difficulty multitasking, and increased fear of income and job losses [18]. Moreover, it can cause profound psychosocial effects among individuals [19].

Considering the long-term effects of brain fog on psychosocial, physiological, and socioeconomic levels, it is important to evaluate this condition in society. No standard diagnostic criteria have been established for this condition [1]. In response to this need, Debowska et al. [20] developed the 23-item Brain Fog Scale (BFS). The validity of this scale was confirmed in 1452 university students, and it was found to have good psychometric properties. However, in Türkiye, there is an inadequacy in this regard. Therefore, we hypothesized that the BFS is a valid and reliable scale for evaluating brain fog in Türkiye.

### Aims and Goals

In this study, we aimed to evaluate the reliability and validity of the BFS in a Turkish population as well as its convergent validity with the Healthy and Unhealthy Eating Behavior Scale (HUEBS) and brain fog status in the same population.

## 2. Materials and Methods

This study was conducted using a survey prepared online on Google Forms, and the data collection process was carried out between January and February 2024. The scale was first applied to a pilot population to make the necessary design corrections, and then principal component analysis and confirmatory factor analysis (CFA) were conducted in the new sample. The literature recommends that the sample size be at least 5–10 times the number of scale items [21]. Accordingly, the sample size for the 23-item scale was calculated to be 115. Different samples were collected for the pilot study (Study 1) and Study 2. During the data collection, 125 participants were reached for the pilot study and 573 for Study 2. The sample for Study 2 was divided into two different data sets for exploratory factor analysis (EFA) and confirmatory factor analysis. Participants who were aged 18 years or older and volunteered to participate in the study were included. Before starting the study, ethical approval was obtained from Acibadem Mehmet Ali Aydınlar University Medical Research Ethics Committee with decision number 2024-5/184. In addition, the participants were asked to sign a consent form. The Declaration of Helsinki’s principles were followed in the conduct of the study. The data collection form used in the study consisted of three sections. The first section included questions regarding sociodemographic characteristics and general eating habits. BFS and HUEBS were used in the second and third sections, respectively.

### 2.1. Physical Activity

To obtain information regarding physical activity status, the participants were asked how many days they performed 30 min of exercise that was sufficient to increase their respiratory rate in the last week. The participants answered this question on a scale ranging from “I did it for 1 day” to “I did it for 7 days”.

### 2.2. Perception of Body Weight

The participants were asked the question “How do you evaluate your body weight?” to assess their perceptions of their body weight. They chose one of the following options: underweight, normal, overweight, or obese.

### 2.3. General Health Status

The participants were asked how they evaluated their general health status [22]. They answered this question on a 5-point Likert scale ranging from “poor” to “excellent”.

### 2.4. Previous COVID-19 Diagnosis Status

The participants were asked about their previous COVID-19 diagnosis, and their brain fog status was evaluated accordingly.

### 2.5. Brain Fog Scale

The BFS was developed in 2024 by Debowska et al. [20]. The validity and reliability of the scale was assessed in a study including 1452 university students. The scale was reported to have good reliability (Cronbach’s alpha values: Factor 1 = 0.79, Factor 2 = 0.80, Factor 3 = 0.78). Furthermore, a three-factor structure, including mental fatigue, impaired cognitive acuity, and confusion, was confirmed. The mental fatigue factor has six items; the impaired cognitive acuity factor (nine items) and the confusion factor (eight items). The scale is scored on a 5-point Likert scale ranging from “never” to “always” and is based on the scores obtained from the subscale. In line with this information, the BFS is a valid scale with good psychometric properties for the assessment of brain fog in clinical and research settings [20].

#### Language and Cultural Adaptation of the Brain Fog Scale

Permission to use the scale was obtained from the scale developer study group in November 2023. Language adaptation of the scale was performed using backtranslation techniques. Using a standardized process suggested by Brislin [23], a bilingual researcher translated the scale items from English to the target language. Then, the translated scale was retranslated by different bilingual researchers. All errors and inconsistencies were highlighted, and the backtranslation comparison process was continued until the inconsistencies were eliminated, as suggested by Bracken and Barona [24].

### 2.6. Healthy and Unhealthy Eating Behavior Scale

The HUEBS was developed by Guertin et al. [25]. The items of this scale were formulated considering the recommendations of Canada’s Food Guide [26]. The scale has 2 subscales measuring healthy and unhealthy eating behaviors, each consisting of 11 items. The participants were asked how much they consume different food categories representing both eating behaviors. Examples of healthy eating behavior items are “I eat fruit” and “I eat vegetables,” and an example of unhealthy eating behavior item is “I use white sugar or artificial sweeteners”. These items are scored on a 7-point scale ranging from “never” to “always” [25]. The validity and reliability of the Turkish version of the HUEBS have not yet been evaluated. Before starting this study, the validity and reliability of the scale were evaluated and the 19-item and 2-factor structure was confirmed.

### 2.7. Statistical Analysis

For categorical variables, descriptive statistics were expressed as frequency and percentage, and for numerical variables, as mean ± standard deviation ($\bar{X}\;$± SD), median, minimum, and maximum values. The Shapiro–Wilk test was used to assess the suitability of numerical variables for normal distribution. Cronbach’s alpha coefficient was used to evaluate the reliability based on item variances, and if >0.80, the test was considered to have high reliability [27]. Furthermore, the Kaiser–Mayer–Olkin (KMO) test was employed to evaluate the adequacy of the sample size and Bartlett’s sphericity test to evaluate the suitability of the scale for factor analysis. Exploratory factor analysis and principal component analysis were employed to determine the factor structure of the scale. For the construct and component validity of the scale factors, confirmatory factor analysis and the Varimax rotation technique as a factor retention method were used. The suitability of the model realized using the maximum likelihood technique was tested with the root mean square error approximation (RMSEA ≤ 0.08), comparative fit index (0.90 ≤ CFI ≤ 1.00), normed fit index (0.90 ≤ NFI ≤ 1.00), goodness-of-fit index (0.85 ≤ GFI ≤ 1.00), adjusted goodness-of-fit index (0.85 ≤ AGFI ≤ 1.00), and standardized root mean square residual (SRMR < 0.10). For the construct validity of the scale, convergent validity was taken into consideration. For convergent validity, all the composite reliability (CR) values are expected to be greater than the AVE values, and the AVE values are expected to be greater than 0.5. Furthermore, the standardized factor loadings of the items should be greater than 0.5, and the CR value should be greater than 0.7 [28]. The relationships between the scales were analyzed using Pearson’s correlation coefficient. Correlation coefficient < 0.2 indicates an extremely weak correlation; 0.2–0.4, weak correlation; 0.4–0.6, moderate correlation; 0.6–0.8, high correlation; and >0.8, extremely high correlation [29]. Statistical analyses were conducted using SPSS version 27 [30] and R Project version 3.6.1 [31].

## 3. Results

### 3.1. Pilot Study—Study 1

In the pilot study, 78.4% and 21.6% of the participants (*n* = 125) were women and men, respectively. Their mean age was 35.52 ± 7.59 years (Appendix A). In the pilot study, the Cronbach’s Alpha value of the BFS was 0.966 and the scale was found to have high reliability. According to the results of the item analysis, it was determined that the Cronbach’s alpha values of 23 scale items were between 0.963 and 0.966, and since there were no items with a value below 0.30, there was no need to remove any item from the scale and the BFS was suitable for validity analysis (unshowned data).

### 3.2. Exploratory Factor Analysis and Confirmatory Factor Analysis—Study 2

#### 3.2.1. Descriptive Statistics of Study 2

The mean age of the participants included in the exploratory factor analysis was 38.60 ± 10.13 years. Of these participants, 42.6% were overweight, 54.8% had good health, and 59.1% had previously been diagnosed with COVID-19.

The mean age of the participants included in the confirmatory factor analysis was 37.12 ± 8.45 years. Of these participants, 45.8% had normal weight, 58.3% had good health, and 64.7% had previously been diagnosed with COVID-19 (Table 1).

#### 3.2.2. Cronbach’s Alpha Values and Kaiser-Meyer-Olkin Test Results of the Brain Fog Scale

The Cronbach’s alpha value of the BFS, which consists of 23 items, was 0.966, indicating that the scale has high reliability. Cronbach’s alpha values ranged between 0.963 and 0.966 when selected items were deleted, and it was determined that it was unnecessary to remove items from the scale (Table 2). The KMO value was 0.951. The Bartlett sphericity test chi-squared value was significant at χ^2^ = 6177.082 and *p* < 0.001. These findings indicate that the dataset was suitable for exploratory factor analysis (Table 3).

#### 3.2.3. Exploratory Factor Analysis

Exploratory factor analysis revealed a three-factor structure with factor loadings >0.30 and eigenvalues >1, explaining 78.485% of the total variance. In the principal component analysis, it was decided not to remove any items from the scale because there was no item with a factor loading of <0.30 and no item with a difference of at least 0.10 between two factor loadings with low loading values. The resulting factors were named “mental fatigue,” “impaired cognitive acuity,” and “confusion”. In line with these results, it was determined that BFS has a five-point Likert-type, 23-item, and three-factor structure and that the total score to be obtained from the scale ranges from 0 to 92 (Table 4).

#### 3.2.4. Confirmatory Factor Analysis

Confirmatory factor analysis was employed to confirm the three-factor structure of the BFS revealed by the exploratory factor analysis. In the first model provided by the confirmatory factor analysis, the criterion values were met, and it was unnecessary to remove items from the scale as there were no items with a factor loading <0.3 (Figure 1).

The fit values of the model obtained by the confirmatory factor analysis to the structural equation model are shown in Table 5. Because the data obtained were within the threshold values, it was determined that the model had a good fit index.

Because the standardized factor loadings of the items were between 0.606 and 0.922, the CR values were greater than 0.7, and the average variance extracted values were greater than 0.5, it was determined that all three structures have convergent validity (Table 6).

#### 3.2.5. Findings Related to Brain Fog Scale

It was determined that the BFS total score varied between 0 and 92 and its mean was 32.48 ±
17.62. The mean values of the mental fatigue, impaired cognitive acuity, and confusion subscales were 12.31 ± 9.33, 7.28 ± 7.35, and 12.88 ± 5.14, respectively (Table 7).

A statistically significant difference (*p* < 0.001) was observed between the subscale and total scores of the BFS according to the COVID-19 diagnosis status of the individuals (Table 8).

An extremely weak positive (r = 0.123; *p* < 0.05) correlation was observed between the BFS total scores and HUEBS total scores. In addition, as the individuals’ BFS total scores increased, a 12.3% increase in their HUEBS total scores was found (Table 9).

## 4. Discussion

Although numerous disorders are frequently accompanied by brain fog, there is currently no specific test to detect this pathology (2). In this direction, we aimed to evaluate the reliability and validity of the BFS in a Turkish population considering this inadequacy in Türkiye. We first conducted a pilot study to make the necessary design corrections. The Cronbach’s alpha values of the 23-item BFS indicated good reliability [32] and were suitable for validity analysis without the need to remove any items from the scale.

After the pilot study, we conducted Study 2. The Cronbach’s alpha coefficient of the scale, which consists of 23 items, was found to be 0.966. It was observed that if the selected item was deleted, the coefficients changed between 0.963 and 0.966. The alpha value is an important index that indicates test reliability [33]. For this value, ≥0.9 indicates excellent reliability; 0.9–0.8, good reliability; and 0.8–0.7, acceptable reliability [34]. It was reported that the Cronbach’s alpha values of the three factors of the original scale developed by Debowska et al. [20] were 0.79, 0.80, and 0.78, respectively, and indicated good reliability. These findings, similar to the original scale, point to the reliability of our scale and that the scale items are related to each other as a whole. Furthermore, we found the KMO value to be 0.951. A KMO value greater than 0.70 is considered to indicate good acceptability [35]. This value shows that our sample is sufficient for factor analysis.

It is reported in the literature that the minimum acceptable value for factor loading is 0.30 [36]. The factor loadings of the scale items ranged between 0.637 and 0.905. In this direction, the factor loadings of the scale meet the minimum level. The exploratory factor analysis revealed a three-factor structure explaining 78.485% of the total variance. These factors are mental fatigue, impaired cognitive acuity, and confusion. The confirmatory factor analysis confirmed this three-factor structure with 23 items. Similarly, this construct was confirmed on the original scale, and it was shown that Factor 1 explained 47.76%, Factor 2 explained 6.67%, and Factor 3 explained 6.22% of the variance [20]. These findings suggest that the BFS is a valid tool for the assessment of mental fatigue, which can affect a person’s mental performance; impaired cognitive acuity, characterized by difficulty in thinking, learning, and concentrating; and confusion, defined as a feeling of detachment from one’s surroundings. Additionally, the scoring of the scale is based on the scores obtained from these subscales.

Notably, eating behavior and habits can be effective in stabilizing or improving cognitive functions [37]. A study evaluating brain fog in perimenopausal women reported that cognitive status was different in participants following different diets [38]. This finding suggests that brain fog is influenced by eating behaviors. In parallel with this, we found that as the BFS total score increased, the HUEBS total score also increased, indicating that brain fog and eating behavior are associated.

Considering previous studies, we hypothesized that brain fog scores may be higher in people previously diagnosed with COVID-19 [16,39,40,41,42]. Related to this, it has been identified that sustained systemic inflammation and persistent localized blood–brain barrier dysfunction in COVID-19 survivors may predispose to brain fog in the long term [43]. A systematic review of 17 studies reporting 41.249 long-COVID patients found that the combined prevalence of mental health problems and brain fog was 20.4% at all time points (3–24 months), and that the prevalence was lower among previously hospitalized patients than among community-managed patients [44]. Debowska et al. [20] also reported that participants previously diagnosed with COVID-19 experienced more symptoms of mental fatigue, impaired cognitive acuity, and confusion. We demonstrated that the subscale scores of mental fatigue, impaired cognitive acuity, and confusion were higher in participants diagnosed with COVID-19. This finding, similar to the literature, indicates that brain fog may be a long-term symptom of COVID-19. Accordingly, the BFS may be a useful tool to characterize increased brain fog after COVID-19. However, it should not be ignored that this scale has been validated in the general population and it should be considered to evaluate the validity and reliability in specific patient groups.

This study has some limitations. First, the data of the questionnaire were collected online. In other words, all data were based on the participants’ statements. In particular, it is likely that participants who reported that they had been diagnosed with COVID-19 did so without relying on the test result. Conversely, it is possible that participants who stated that they were not diagnosed may actually be asymptomatic COVID-19 cases. Secondly, there was an imbalance among participants in terms of gender, age, and educational status. It should be noted that different constructs may be validated for participants of different gender, age, and educational status. Further studies examining them in a homogeneous way are needed to make more specific analyses.

## 5. Conclusions

To the best of our knowledge, this is the first study to test the validity and reliability of the BFS developed by Debowska et al. [20] in a Turkish population. We confirmed the validity and reliability of the 23 items and three factors of this scale: mental fatigue, impaired cognitive acuity, and confusion. We also demonstrated that the brain fog scores were higher in participants previously diagnosed with COVID-19, indicating that brain fog may be a long-term symptom of COVID-19. Accordingly, the 23-item BFS was found to be a valid and reliable tool for the Turkish population. Further studies are warranted to determine the cutoff points.

## Figures and Tables

**Figure 1 ijerph-21-00774-f001:**
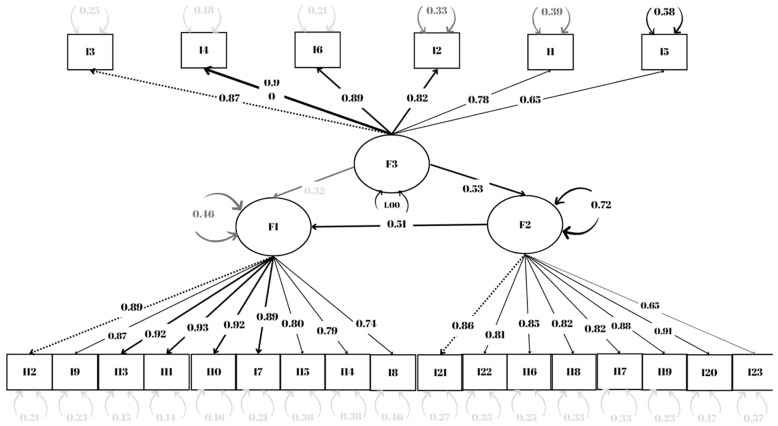
Confirmatory factor analysis model of the study.

**Table 1 ijerph-21-00774-t001:** Descriptive statistics of the demographic, health, and nutritional findings of individuals included in Study 2.

	Dataset 1 (AFA)						Dataset 2 (CFA)					
	Man (*n* = 40)		Woman (*n* = 190)		Total (*n* = 230)		Man (*n* = 67)		Woman (*n* = 276)		Total (*n* = 343)	
	*n*	%	*n*	%	*n*	%	*n*	%	*n*	%	*n*	%
Age (year) ($\bar{X}\pm SD$)	36.80 ± 11.11		38.98 ± 9.90		38.60 ± 10.13		37.34 ± 8.79		37.07 ± 8.38		37.12 ± 8.45	
Educational Level												
Primary school	1	2.5	8	4.2	9	3.9	2	3.0	2	0.7	4	1.2
High school	3	7.5	21	11.1	24	10.4	8	11.9	28	10.1	36	10.5
Bachelor degree	21	52.5	110	57.9	131	57.0	39	58.2	160	58.0	199	58.0
MSc and PhD	15	37.5	51	26.8	66	28.7	18	26.9	86	31.2	104	30.3
Occupation												
Civil servant	15	37.5	63	33.2	78	33.9	19	28.4	77	27.9	96	28.0
Private sector	12	30.0	57	30.0	69	30.0	23	34.3	112	40.6	135	39.4
Self-employment	5	12.5	19	10.0	24	10.4	16	23.9	32	11.6	48	14.0
Retired	0	0.0	12	6.3	12	5.2	2	3.0	8	2.9	10	2.9
Housewife	0	0.0	5	2.6	5	2.2	0	0.0	2	0.7	2	0.6
Student	4	10.0	7	3.7	11	4.8	5	7.5	5	1.8	10	2.9
Unemployed	4	10.0	27	14.2	31	13.5	2	3.0	40	14.5	42	12.2
Chronic Disease Diagnosed by a Doctor												
Yes	12	30.0	96	50.5	108	47.0	22	32.8	122	44.2	144	42.0
No	28	70.0	94	49.5	122	53.0	45	67.2	154	55.8	199	58.0
Skipping Meals												
I don’t skip meals	2	5.0	40	21.1	42	18.3	11	16.4	48	17.4	59	17.2
Sometimes I skip meals	23	57.5	86	45.3	109	47.4	32	47.8	136	49.3	168	49.0
I skip breakfast	9	22.5	32	16.8	41	17.8	16	23.9	34	12.3	50	14.6
I skip lunch	6	15.0	28	14.7	34	14.8	6	9.0	57	20.7	63	18.4
I skip dinner	0	0.0	4	2.1	4	1.7	2	3.0	1	0.4	3	0.9
BMI Classification												
Underweight	3	7.5	6	3.2	9	3.9	2	3.0	11	4.0	13	3.8
Normal	15	37.5	91	47.9	106	46.1	24	35.8	158	57.2	182	53.1
Overweight	16	40.0	48	25.3	64	27.8	26	38.8	67	24.3	93	27.1
Obese	6	15.0	45	23.7	51	22.2	15	22.4	40	14.5	55	16.0
BMI (kg/m^2^) ($\bar{X}\pm SD$)	25.48 ± 4.49		26.31 ± 5.64		26.16 ± 5.46		26.65 ± 4.00		24.86 ± 4.74		25.21 ± 4.66	
Thinking of Eating Healthy												
Yes	22	55.0	105	55.3	127	55.2	36	53.7	155	56.2	191	55.7
No	18	45.0	85	44.7	103	44.8	31	46.3	121	43.8	152	44.3
Physical Activity Status in the Last Week												
I did it for 1 day	18	45.0	94	49.5	112	48.7	35	52.2	148	53.6	183	53.4
I did it for 2 days	12	30.0	48	25.3	60	26.1	16	23.9	59	21.4	75	21.9
I did it for 3 days	5	12.5	22	11.6	27	11.7	6	9.0	37	13.4	43	12.5
I did it for 4 days	2	5.0	11	5.8	13	5.7	5	7.5	15	5.4	20	5.8
I did it for 5 days	0	0.0	12	6.3	12	5.2	4	6.0	10	3.6	14	4.1
I did it for 6 days	3	7.5	3	1.6	6	2.6	1	1.5	5	1.8	6	1.7
I did it for 7 days	0	0.0	0	0.0	0	0.0	0	0.0	2	0.7	2	0.6
Body Weight Assessment Status												
Underweight	3	7.5	8	4.2	11	4.8	1	1.5	5	1.8	6	1.7
Normal	15	37.5	70	36.8	85	37.0	33	49.3	124	44.9	157	45.8
Overweight	19	47.5	79	41.6	98	42.6	24	35.8	113	40.9	137	39.9
Obese	3	7.5	33	17.4	36	15.7	9	13.4	34	12.3	43	12.5
General Health Evaluation Status												
Poor	0	0	9	4.7	9	3.9	2	3.0	12	4.3	14	4.1
Fair	14	35.0	58	30.5	72	31.3	13	19.4	62	22.5	75	21.9
Good	21	52.5	105	55.3	126	54.8	37	55.2	163	59.1	200	58.3
Very good	5	12.5	16	8.4	21	9.1	15	22.4	38	13.8	53	15.5
Excellent	0	0.0	2	1.1	2	0.9	0	0.0	1	0.4	1	0.3
COVID-19 Diagnosis Status												
Yes	29	72.5	107	56.3	136	59.1	37	55.2	185	67.0	222	64.7
No	11	27.5	83	43.7	94	40.9	30	44.8	91	33.0	121	35.3

**Table 2 ijerph-21-00774-t002:** Cronbach’s alpha coefficients if item deleted.

Item	Cronbach’s Alpha Coefficients If Item Deleted
I1	0.965
I2	0.965
I3	0.965
I4	0.965
I5	0.966
I6	0.966
I7	0.964
I8	0.964
I9	0.964
I10	0.963
I11	0.963
I12	0.963
I13	0.963
I14	0.964
I15	0.964
I16	0.964
I17	0.964
I18	0.964
I19	0.964
I20	0.964
I21	0.964
I22	0.965
I23	0.965
Total	0.966

I: item.

**Table 3 ijerph-21-00774-t003:** KMO, and Bartlett’s test findings of BFS.

KMO and Bartlett’s Test		
Kaiser–Meyer–Olkin Measure of Sampling Adequacy		0.951
Bartlett’s Test of Sphericity	Approx. chi-square	6177.082
	df	253
	Sig.	<0.001 ***

KMO: Kaiser–Meyer–Olkin, *** *p* < 0.001.

**Table 4 ijerph-21-00774-t004:** Dimensions of the scale obtained as a result of the exploratory factor analysis.

Items	English and Turkish Versions of the Items	Mental Fatigue	Impaired Cognitive Acuity	Confusion
I11	I couldn’t think clearly./Net şekilde düşünemiyordum.	0.886		
I13	I have found it difficult to organise my thoughts./Düşüncelerimi organize etmekte zorlandım.	0.885		
I10	I have found it difficult to concentrate./Konsantre olmakta zorlandım.	0.880		
I12	I have had a hard time finding the right words./Doğru kelimeleri bulmakta zorlandım.	0.874		
I9	I have found it difficult to think logically./Mantıklı düşünmekte zorlandım.	0.872		
I7	I have found it difficult to remember and understand new information./Yeni bilgileri hatırlamakta ve anlamakta zorlandım.	0.854		
I14	I have felt like my mind’s gone blank./Zihnimin boşaldığını hissettim.	0.797		
I15	I have found it difficult to understand words when reading./Okurken kelimeleri anlamakta zorlandım.	0.795		
I8	I have found myself forgetting certain words, such as the names of objects./Kendimi nesnelerin isimleri gibi belirli kelimeleri unuturken buldum.	0.725		
I21	I have felt lost./Kendimi kaybolmuş hissettim.		0.858	
I18	I have felt spacey./Gerçeklikten koptuğumu hissettim.		0.842	
I22	I have felt absent, as if I were living in my own world./Sanki kendi dünyamda yaşıyormuşum gibi bir yokluk hissi yaşadım.		0.835	
I19	I have felt confused./Kafamın karıştığını hissettim.		0.800	
I17	I have been daydreaming./Hayal dünyasında gibiydim.		0.798	
I16	I have had a hard time understanding what others say./Başkalarının ne dediğini anlamakta zorlandım.		0.740	
I20	I have experienced thought blocking./Bazen zihni takılmalar yaşadım.		0.733	
I23	My thoughts have been moving quickly./Düşüncelerim beynimde yarışıyordu.		0.720	
I3	I have felt fatigued./Yorgun hissettim.			0.905
I6	I have felt sleepy./Uyuşmuş/uykulu gibiydim.			0.836
I4	I have been easily distracted./Kolayca dikkatim dağıldı.			0.798
I2	I have felt mentally exhausted./Zihinsel olarak yorgun olduğumu hissettim.			0.784
I1	My thinking has been slow./Düşüncelerimin yavaşladığını hissettim.			0.774
I5	I have found myself getting annoyed./Kendimi sinirlenirken buldum.			0.637
EVR		**57.306**	**11.430**	**9.750**
EV		**13.180**	**2.629**	**2.242**
a		**0.974**	**0.955**	**0.920**

I: item; EVR: explained variance rate; EV: eigenvalue; a: Cronbach’s alpha coefficient.

**Table 5 ijerph-21-00774-t005:** Fit values of the scale.

Fit Index	Threshold Values	Analysis Results
Degree of Freedom	-	227
Chi-squared/sd	0 ≤ Chi-squared/sd ≤ 2	0.663
RMSEA	RMSEA ≤ 0.08	0.011
NFI	0.90 ≤ NFI ≤ 1.00	0.991
CFI	0.90 ≤ CFI ≤ 1.00	0.998
SRMR	SRMR < 0.08	0.047
GFI	0.85 ≤ GFI ≤ 1.00	0.993
AGFI	0.85 ≤ AGFI ≤ 1.00	0.992

AGFI: adjusted goodness-of-fit index; CFI: comparative fit index; GFI: goodness-of-fit index; NFI: normed fit index; RMSEA: root mean square error; SRMR: standardized root mean square residual.

**Table 6 ijerph-21-00774-t006:** Component values of factors and items as a result of confirmatory factor analysis.

Factors and Items	SFL > 0.5	CR > 0.7	AVE > 0.4/0.5	Cronbach’s α > 0.7
**Mental Fatigue**		**0.940**	**0.638**	**0.961**
I7	0.795			
I8	0.606			
I9	0.866			
I10	0.830			
I11	0.844			
I12	0.867			
I13	0.859			
I14	0.735			
I15	0.751			
**Impaired Cognitive Acuity**		**0.927**	**0.614**	**0.945**
I16	0.819			
I17	0.797			
I18	0.805			
I19	0.775			
I20	0.711			
I21	0.847			
I22	0.843			
I23	0.650			
**Confusion**		**0.914**	**0.642**	**0.923**
I1	0.769			
I2	0.793			
I3	0.922			
I4	0.829			
I5	0.669			
I6	0.803			

AVE: average variance extracted; CR: composite reliability; I: item; SFL: standardized factor loading.

**Table 7 ijerph-21-00774-t007:** Statistics of BFS factors and total scores.

Factor	*n*	$\bar{X}\pm SD$	Median (Min–Max)
Mental Fatigue	343	12.31 ± 9.33	10 (0–36)
Impaired Cognitive Acuity	343	7.28 ± 7.35	5 (0–32)
Confusion	343	12.88 ± 5.14	12 (0–24)
Total BFS	343	32.48 ± 17.62	30 (0–92)

BFS: Brain Fog Scale.

**Table 8 ijerph-21-00774-t008:** Comparison of the subscale and total scores of the BFS according to the COVID-19 diagnosis status of the individuals.

	COVID-19 Diagnosis Status	*n*	$\bar{X}\pm SD$	Median (Min–Max)	t	*p*
Mental Fatigue	Yes	178	15.65 ± 9.61	17 (0–36)	7436	<0.001
	No	165	8.69 ± 7.53	9 (0–36)		
Impaired Cognitive Acuity	Yes	178	9.85 ± 7.85	8 (0–32)	7213	<0.001
	No	165	4.51 ± 5.61	2 (0–32)		
Confusion	Yes	178	14.47 ± 4.58	13 (0–24)	6266	<0.001
	No	165	11.16 ± 5.17	11 (0–24)		
Total BFS	Yes	178	34.75 ± 18.64	38 (0–92)	9137	<0.001
	No	165	35.20 ± 17.50	22 (0–92)		

BFS: Brain Fog Scale.

**Table 9 ijerph-21-00774-t009:** Correlation Coefficient Between the BFS and HUEBS.

		Mental Fatigue	Impaired Cognitive Acuity	Confusion	Total BFS
Impaired Cognitive Acuity	r	0.651	1		
	*p*	<0.001 ***			
Confusion	r	0.562	0.495	1	
	*p*	<0.001 ***	<0.001 ***		
Total BFS	r	0.898	0.867	0.758	1
	*p*	<0.001 ***	<0.001 ***	<0.001 ***	
Total HUEBS	r	0.101	0.078	0.152	0.123
	*p*	0.063	0.151	0.005 **	0.023 *

BFS: Brain Fog Scale; HUEBS: Healthy and Unhealthy Eating Behavior Scale; r: Pearson’s correlation coefficient; * *p* < 0.05; ** *p* < 0.01; *** *p* < 0.001.

## Data Availability

The data presented in this study are available on request from the corresponding author due to privacy.

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
