# Peer review of "Adaptation and Validation of the Turkish Version of the Brain Fog Scale"

_ijerph, 2024, doi:10.3390/ijerph21060774_

Round 1
Reviewer 1 Report
Comments and Suggestions for Authors
Check Table 2, mistakenly the decimal point is replaced with a comma, make sure all the entries in Table 2 are correct.
Author Response
Dear Reviewer,
We have revised our manuscript taking your suggestions into account. Thank you very much for all your contributions.

Reviewer 2 Report
Comments and Suggestions for Authors
1. table 1 seems to be 3 pages long. maybe it'll be better to repeat header rows on thse 3 pages. Same for table 4
2. all the numbers seems satisfactory but I'm not sure about the relation between covid and BFS. maybe I'm missing the related text but please explain better
Comments on the Quality of English Language.
Author Response
Dear reviewer,
We have revised our article taking your suggestions into account. Thank you very much for all your contributions.

Reviewer 3 Report
Comments and Suggestions for Authors
Brain fog is a condition that is characterized by poor concentration, memory loss, decreased cognitive function and mental fatigue. The authors evaluated the reliability and validity of the Brain Fog Scale developed by Debowska et al. in a Turkish population. They conducted a pilot study (n=125) to confirm the suitability of the scale for validity analyses, and then conducted exploratory (n=230) and confirmatory factor analyses (n=343). Results of analyses indicated that the 23-item, 3-factor (mental fatigue, impaired cognitive acuity, confusion) Brain Fog Scale is valid and reliable. They demonstrated that the brain fog scores were higher in participants previously diagnosed with COVID-19, indicating that brain fog may be a long-term symptom of COVID-19. They also reported an association between brain fog and eating behavior. In general, the study provides an assessment of brain fog for the Turkish population. However, there are some shortcomings in the paper as noted below.
1. The reliability and validity of the Brain Fog Scale had been investigated and elucidated in previous study (Debowska et al). The authors of the current study are strongly encouraged to list the specific novel approaches and/or findings from their current study in the Abstract and discuss their specific clinical implications and applications in the Discussion.
2. Data for this study was obtained from online questionnaire rather than face-to-face questionnaire. Therefore, the reliability of the data has yet to be verified.
3. Participation in the study was predominately female, young, and highly educated. The adaptability of the scale for generalization to popular groups needs to be further assessed.
Author Response

(The authors gave the same response as above.)
